Comparison of lodgepole and jack pine resin chemistry: implications for range expansion by the mountain pine beetle, Dendroctonus ponderosae (Coleoptera: Curculionidae)

Clark Erin L. 1 eclark1@unbc.ca
Pitt Caitlin 1
Carroll Allan L. 2
Lindgren B. Staffan 1
Huber Dezene P.W. 1
1 Ecosystem Science and Management Program, University of Northern British Columbia , Prince George , British Columbia , Canada
2 Department of Forest & Conservation Sciences, University of British Columbia , Vancouver , British Columbia , Canada
Higley Leon
Electronic publication date: 2014 Feb 11
Publication date: 2014
Volume: 2
Electronic Location ID: e240
Received 2013 Oct 9; Accepted 2013 Dec 17
Copyright: © 2014 Clark et al.
Copyright year: 2014
Copyright holder: Clark et al.
License: This is an open access article distributed under the terms of the Creative Commons Attribution License, which permits unrestricted use, distribution, and reproduction in any medium, provided the original author and source are credited.
License URL: https://creativecommons.org/licenses/by/3.0/

Keywords: Secondary metabolites, Terpenes, Coleoptera, Curculionidae, Scolytinae, Plant defense, Bark beetles

Funding: Funding for this research was generously provided by the Natural Resources Canada Mountain Pine Beetle Initiative research grant (8.45), the Natural Sciences and Engineering Research Council of Canada, the Canada Research Chairs Program, the Canada Foundation for Innovation, and the British Columbia Knowledge Development Fund. The funders had no role in study design, data collection and analysis, decision to publish, or preparation of the manuscript.

==============================
The mountain pine beetle, Dendroctonus ponderosae, is a significant pest of lodgepole pine in British Columbia (BC), where it has recently reached an unprecedented outbreak level. Although it is native to western North America, the beetle can now be viewed as a native invasive because for the first time in recorded history it has begun to reproduce in native jack pine stands within the North American boreal forest. The ability of jack pine trees to defend themselves against mass attack and their suitability for brood success will play a major role in the success of this insect in a putatively new geographic range and host. Lodgepole and jack pine were sampled along a transect extending from the beetle’s historic range (central BC) to the newly invaded area east of the Rocky Mountains in north-central Alberta (AB) in Canada for constitutive phloem resin terpene levels. In addition, two populations of lodgepole pine (BC) and one population of jack pine (AB) were sampled for levels of induced phloem terpenes. Phloem resin terpenes were identified and quantified using gas chromatography. Significant differences were found in constitutive levels of terpenes between the two species of pine. Constitutive α-pinene levels – a precursor in the biosynthesis of components of the aggregation and antiaggregation pheromones of mountain pine beetle – were significantly higher in jack pine. However, lower constitutive levels of compounds known to be toxic to bark beetles, e.g., 3-carene, in jack pine suggests that this species could be poorly defended. Differences in wounding-induced responses for phloem accumulation of five major terpenes were found between the two populations of lodgepole pine and between lodgepole and jack pine. The mountain pine beetle will face a different constitutive and induced phloem resin terpene environment when locating and colonizing jack pine in its new geographic range, and this may play a significant role in the ability of the insect to persist in this new host.

Introduction

The primary host of the mountain pine beetle, Dendroctonus ponderosae Hopkins (Coleoptera: Curculionidae) in its native range in western Canada is lodgepole pine (Pinus contorta Dougl. var. latifolia Engelm.), but the insect is also capable of utilizing other species of pine, including jack pine (P. banksiana Lamb.) (Furniss & Schenk, 1969; Safranyik & Linton, 1982; Cerezke, 1995; Cullingham et al., 2011; Erbilgin et al., 2013). Lodgepole pine is found throughout northwestern North America, but occurs primarily in British Columbia (BC) and north-central Alberta, Canada. The northeastern portion of the range of lodgepole pine is contiguous with jack pine in Alberta. Jack pine extends east across Canada and into the northeastern United States (Little, 1971). Where the two species’ ranges overlap, they form a hybrid zone (Moss, 1949), which in recent years has been successfully invaded by the mountain pine beetle (Langor, Rice & Williams, 2007). More recently, beetles have been found successfully reproducing in pure jack pine stands (Cullingham et al., 2011). Historically, mountain pine beetle populations have been most common west of the Rocky Mountains (Safranyik & Carroll, 2006). Non-forested prairies, the high elevations of the mountains, and cold winters at higher latitudes have contributed to confining it to that distribution. Its recent invasion of the pine forests east of the Rocky Mountains in north-central Alberta (Langor, Rice & Williams, 2007; Cullingham et al., 2011) has raised concerns that it may have the capacity to spread eastward through Canada’s extensive jack pine forest (Logan & Powell, 2001; Safranyik et al., 2010).

Invasion of a novel habitat by either a native or an alien herbivore may require that animal to exploit new host-plant populations or species. Release from competition and/or predation associated with the herbivore’s native habitat are other factors that may contribute to the success of an invasive species (Keane & Crawley, 2002). Encountering plants that are not able to defend themselves against specific forms of herbivory often enables an invading species to do well in a new environment. For instance, the emerald ash borer, Agrilus planipennis Fairmaire (Coleoptera: Buprestidae), an insect native to Asia where it is not considered a major pest, is very destructive to several species of ash (Fraxinus spp.) in North America (Haack et al., 2002; Poland & McCullough, 2006). Asian ash species are more resistant to the emerald ash borer (Rebek, Herms & Smitley, 2008), possibly due to significant differences in constitutive phloem chemistry, including compounds that are toxic or deterrent to other herbivores (Eyles et al., 2007).

In a similar fashion, mountain pine beetle invading previously unoccupied range may encounter hosts not adapted to defending against them, i.e., they enter relatively ‘defense-free space’ (Ghandi & Herms, 2010). Cudmore et al. (2010) found that the productivity of beetles was higher in tree populations putatively naïve to mountain pine beetle outbreaks, but did not investigate potential mechanisms for this phenomenon. In northern Alberta, recent evidence for successful reproduction in jack pine (Cullingham et al., 2011) suggests the mountain pine beetle may be successful in these novel hosts, but the fitness and resultant dynamics of beetle populations in this new environment is presently unknown. Factors such as climate, the abundance and distribution of susceptible and suitable host trees, and overall stand structure may play key roles in determining the ultimate outcome of the spread into the jack pine forests. Since the susceptibility and suitability of a potential host tree to the mountain pine beetle is primarily a function of its chemistry (Safranyik & Carroll, 2006) variation in tree chemistry in novel habitats will play a key role in determining the insect’s success.

Among the major defenses used by conifers against attacking organisms are their resin terpenes. Terpenes can serve as attractants or repellants for bark beetles (Gershenzon & Croteau, 1991; Pureswaran & Borden, 2003; Keeling & Bohlmann, 2006) and have been implicated in host selection (Moeck & Simmons, 1991). They have been found to be toxic to bark beetles (Smith, 1961, 1963, 1965; Raffa et al., 1985), precursors to aggregation and anti-aggregation pheromones (Conn et al., 1984), and synergists to pheromones (Miller & Borden, 1990; Borden et al., 1983; Conn et al., 1983). Resins are maintained in most conifers as a primary constitutive defense and produced as an induced defense, triggered when a tree is under attack (Raffa, 1991). For example, lodgepole pine has been found to respond to attack by the mountain pine beetle and its associated fungi with large increases of total terpenes in the tree (Shrimpton, 1973; Raffa & Berryman, 1982, 1983a; Miller, Berryman & Ryan, 1986; Boone et al., 2011). Similarly, in jack pine, total monoterpene concentrations have been found to be elevated in induced tissue compared to levels in constitutive tissue after attack by jack pine budworm (Wallin & Raffa, 1999).

We compared the constitutive resin chemistry and induced responses to simulated attack between lodgepole pine in southern BC (where outbreaks are common) and northern BC (mainly outside of the historical range of the beetle) with those of jack pine in northern and central Alberta. Our objective was to compare the terpene defenses of lodgepole and jack pine trees to gain insight into the beetles’ ability to locate suitable hosts, attract conspecifics, and use the resource to successfully reproduce in these new hosts.

Methods and materials

Constitutive Defenses

Uninfested jack and lodgepole pine trees, as determined by the absence of pitch tubes and frass, were sampled along a transect from the Alberta/Saskatchewan border to Prince George, BC (Fig. 1). We attempted to sample trees at even intervals along the transect, but large gaps where suitable pine could not be found due to agricultural or oil extraction activities, or natural breaks in the forest, could not be avoided. In 2006, seven locations were sampled. In 2007, an additional five sites were sampled to increase the total number of samples. A maximum of ten trees, each with a minimum diameter at breast height (d.b.h.; 1.3 m above the ground) of 15 cm, were sampled per location. At some sites, ten trees meeting the minimum size requirements could not be found so fewer than ten trees were sampled (Fig. 1). A 10 mm diameter punch (No. 149 Arch Punch 10 mm; C.S. Osborne & Co., Harrison, N.J.) was used to remove a disk of bark and phloem at breast height. Each disk was stored in individually labeled envelopes and immediately placed into dry ice where it remained until transferred to a −80°C freezer. Samples were kept at −80°C until they were shipped, buried in dry ice, to the British Columbia Ministry of Forests and Range Forest Research Laboratory, Victoria, BC, for processing and analysis.

Figure 1 Map of sample locations in British Columbia and Alberta.

Phloem samples were processed using gas chromatographic-flame ionization detection analyses (GC-FID) to identify compounds by matching their retention time with synthetic standards. Samples were processed as described in Clark, Carroll & Huber (2010).

Based upon the location of the trees sampled in comparison with the sampling locations by Pollack & Dancik (1985), trees at each location were classified either as lodgepole or jack pine. Terpene concentrations (ppm) and the percent resin content of each terpene were analyzed using a two-sample non-parametric Wilcoxon rank sum test to determine if there were differences between the two species (α = 0.05) as most of the terpene data could not be transformed to meet the assumption of homoscedasticity based upon a Levene’s test. All data were analyzed using R v.2.6.2 (R Development Core Team, 2008). Values below 5 ppm were considered to be zero for analysis.

Enantiomeric composition

To assess the enantiomeric composition of three of the predominant monoterpenes (limonene, α-pinene, and β-pinene), samples from three populations – two lodgepole pine (BC) and one jack pine (Alberta) (Table 1) – taken in the same manner as previously described were analyzed. A Cyclodex-B column (Agilent Technologies) was used with helium as the carrier gas. All data for enantiomeric composition were analyzed using ANOVA with a Tukey’s HSD post-hoc test (α = 0.05). Data were transformed when necessary to meet assumptions of homoscedasticity based upon a visual examination of the residual plots.

Table 1 Sampling locations (Kelowna and Chetwynd – lodgepole pine, Fort McMurray – jack pine), number of trees sampled at each location, and dates of sampling for work on induced defenses.

Location	# of trees	Coordinates	Dates Sampled	
				Day 0	Day 2	Day 14	
Kelowna	10	N 49°57.684′	W 119°42.551′	01 Aug 07	03 Aug 07	15 Aug 07	
Chetwynd	11	N 49°53.145′	W 120°25.121′	05 Aug 07	07 Aug 07	19 Aug 07	
Fort McMurray	10	N 57°21.640′	W 111°32.281′	09 Aug 07	11 Aug 07	23 Aug 07	

Induced Defenses

Uninfested lodgepole pine trees, as determined by an absence of frass and pitch tubes, were selected in a stand near Chetwynd, BC (LP-N), located further north and east than recorded mountain pine beetle infestations prior to 1970 and near the northern edge of the historical range of mountain pine beetle (Safranyik & Carroll, 2006), and near Kelowna, BC (LP-S). In addition, a stand of jack pine – a population assumed to be unexposed to mountain pine beetle – was selected near Fort McMurray, Alberta (JP) (Table 1). Two initial samples were taken by bark punch (10 mm diameter) from both the east and west side of the trees at breast height. Silicone plugs (Mack’s® Pillow Soft® silicone earplugs; McKeon Products, Inc., Warren, MI) were used to cover the wound.

Prior to sampling the tree, the area where the bark punch was to be made was scraped to remove loose debris and then sprayed with ethanol (95%). The tools used in the treatment and sampling process were also sprayed with ethanol before use on each tree to reduce potential contamination. The bark and phloem disks removed from each punch site were saved in individual, labeled envelopes and immediately buried in dry ice in a cooler, where it was kept until it could be transferred to a −80°C freezer for longer-term storage prior to analysis.

Each pine was sampled in the same manner three times in August 2007: (1) the initial sample; (2) two days after initial sampling – with the sampling side randomly selected by a coin flip – approximately 10 mm above the initial punch wound; and (3) fourteen days after initial sampling on the side of the tree opposite from the two-day sample, again approximately 10 mm above the initial punch. The timing of the second sample at two days post-wounding was used because mass attack is normally completed in one or two days following contact of the first mountain pine beetle with the tree (Safranyik & Carroll, 2006). Therefore, it is the initial response by the tree to wounding during this time that is most likely to have an effect on successful colonization.

Samples were shipped to the British Columbia Ministry of Forests and Range Forest Research Laboratory, Victoria, BC for processing using gas chromatographic-flame ionization detection analysis exactly as described previously. One of the two initial phloem samples from each tree (from either east or west side of the tree) was selected by coin toss to be processed for comparison of induced levels of monoterpenes; the other sample was used for analysis of enantiomeric composition. Data on nine monoterpenes (3-carene, limonene, linalool, myrcene, β-phellandrene, α-pinene, β-pinene, pulegone, terpinolene) and the total of all of the 26 terpenes measured were analyzed. Data were analyzed by analysis of variance (ANOVA) to compare monoterpene levels between locations at day 0, 2, and 14. If required, data were transformed [log10(x + 1)] to meet requirements of homoscedasticity based on a visual examination of the residual plots. If the ANOVA was significant (α = 0.05), means were separated by a post-hoc Tukey’s HSD. Data were also analyzed to compare the rate of change in levels of each terpene between the sampling days using the Kruskal-Wallis non-parametric test followed by two sample non-parametric Wilcoxon rank sum test as data could not be transformed to meet requirements of homoscedasticity based on a visual examination of the residual plots.

Results

Constitutive Defenses

Based upon the location of collection, we sampled 50 lodgepole pine trees and 61 jack pine trees. There was no significant difference in tree diameter between the two species in the sample population (P > 0.05). Pollack & Dancik (1985) found that α-pinene and β-phellandrene were the most important variables for differentiating between lodgepole pine and jack pine and the putative hybrid populations in Alberta. With only one exception (which was analyzed as a jack pine), all of the trees sampled at sites considered to be occupied by jack pine had a higher percentage of α-pinene compared with β-phellandrene; and all of the trees from sampling locations occupied by lodgepole pine had a higher percentage of β-phellandrene compared with α-pinene.

There were significant differences (Wilcoxon rank sum test, α = 0.05) between the constitutive concentrations (ppm) in lodgepole and jack pine for all but three of the 26 monoterpenes considered (Table 2). Of the monoterpenes that differed, lodgepole pine had higher levels of 20 of the 23 terpenes with the exceptions of: linalool, pulegone, and α-pinene. Total terpenes were also higher in lodgepole pine than in jack pine. Lodgepole and jack pine also differed significantly in the percentage composition of all but five terpenes in the phloem resin (Table 3). In addition, lodgepole pine had a higher percent composition of all of the terpenes tested that differed between species, except for bornyl acetate (0.37% vs. 1.80%), α-caryophyllene (0.11% vs. 0.29%), pulegone (0.24% vs. 4.16%), and α-pinene (7.09% vs. 58.42%).

Table 2 Mean content of terpenes (PPM) (±1 SE) of lodgepole and jack pine trees.

Terpene	P-valuea	Mean content (ppm)	
		Lodgepole pine	Jack pine	
Borneol	P < 0.001	14.67 ± 3.18	4.18 ± 1.64	
Bornyl acetate	P = 0.004	45.95 ± 12.55	42.52 ± 3.41	
Camphene	P < 0.001	88.47 ± 8.30	30.32 ± 4.51	
Camphor	P = 0.041	6.69 ± 1.93	2.21 ± 0.85	
3-Carene	P < 0.001	1738.14 ± 321.80	226.99 ± 44.95	
α-Caryophyllene	P < 0.001	16.02 ± 3.32	5.47 ± 2.24	
α-Copaene	P < 0.001	13.70 ± 2.68	1.09 ± 0.78	
α-Cubebene	P = 0.273	2.50 ± 1.46	0.79 ± 0.63	
p-Cymene	P < 0.001	32.06 ± 5.11	2.90 ± 1.71	
α-Humulene	P < 0.001	42.35 ± 7.30	3.19 ± 1.42	
Limonene	P < 0.001	848.64 ± 164.35	203.25 ± 48.89	
Linalool	P = 0.220	64.73 ± 9.02	117.74 ± 16.68	
Myrcene	P < 0.001	435.45 ± 45.48	100.18 ± 15.71	
Ocimene	P < 0.001	31.75 ± 8.21	0.00 ± 0.00	
α-Phellandrene	P < 0.001	175.77 ± 17.93	1.62 ± 1.38	
β-Phellandrene	P < 0.001	9096.72 ± 960.59	157.21 ± 40.39	
α-Pinene	P < 0.001	949.81 ± 132.72	2518.14 ± 328.27	
β-Pinene	P < 0.001	921.46 ± 127.02	283.98 ± 43.68	
Pulegone	P = 0.039	30.21 ± 6.81	107.13 ± 22.81	
Sabinene	P < 0.001	141.94 ± 17.74	7.12 ± 2.17	
α-Terpinene	P < 0.001	15.39 ± 2.95	0.00 ± 0.00	
γ-Terpinene	P < 0.001	23.81 ± 5.42	0.00 ± 0.00	
Terpineol	P < 0.001	53.64 ± 8.55	16.50 ± 6.59	
Terpinolene	P < 0.001	352.79 ± 48.44	47.07 ± 13.04	
α-Thujone	P = 0.122	2.25 ± 0.84	1.85 ± 1.20	
Total	P < 0.001	15144.92 ± 1463.47	3881.44 ± 395.26	
a Differences between species as determined by a Wilcoxon rank sum test (α = 0.05, significant differences in bold).

Table 3 Mean relative content of terpenes (%) (±1 SE) of lodgepole and jack pine.

Terpene	P-valuea	Mean relative content (% of total monoterpenes)	
		Lodgepole pine	Jack pine	
Borneol	P = 0.016	0.11 ± 0.02	0.09 ± 0.03	
Bornyl acetate	P < 0.001	0.37 ± 0.10	1.80 ± 0.26	
Camphene	P = 0.874	0.64 ± 0.09	0.60 ± 0.08	
Camphor	P = 0.120	0.06 ± 0.03	0.07 ± 0.03	
3-Carene	P = 0.005	10.56 ± 1.33	8.68 ± 1.48	
α-Caryophyllene	P < 0.001	0.11 ± 0.03	0.29 ± 0.14	
α-Copaene	P < 0.001	0.09 ± 0.02	0.02 ± 0.01	
α-Cubebene	P = 0.307	0.01 ± 0.01	0.04 ± 0.04	
p-Cymene	P < 0.001	0.20 ± 0.04	0.07 ± 0.04	
α-Humulene	P < 0.001	0.32 ± 0.06	0.18 ± 0.12	
Limonene	P = 0.004	5.26 ± 0.85	4.77 ± 0.99	
Linalool	P = 0.090	0.87 ± 0.38	5.93 ± 1.37	
Myrcene	P = 0.005	2.98 ± 0.28	2.16 ± 0.28	
Ocimene	P < 0.001	0.18 ± 0.05	0.00 ± 0.00	
α-Phellandrene	P < 0.001	1.13 ± 0.07	0.02 ± 0.02	
β-Phellandrene	P < 0.001	58.25 ± 1.92	4.19 ± 0.99	
α-Pinene	P < 0.001	7.09 ± 0.91	58.42 ± 2.32	
β-Pinene	P = 0.687	6.74 ± 0.77	6.47 ± 0.72	
Pulegone	P < 0.001	0.24 ± 0.06	4.16 ± 1.23	
Sabinene	P < 0.001	0.91 ± 0.09	0.19 ± 0.06	
α-Terpinene	P < 0.001	0.08 ± 0.02	0.00 ± 0.00	
γ-Terpinene	P < 0.001	0.11 ± 0.02	0.00 ± 0.00	
Terpineol	P < 0.001	0.37 ± 0.06	0.30 ± 0.12	
Terpinolene	P < 0.001	3.29 ± 1.02	1.32 ± 0.35	
α-Thujone	P = 0.128	0.02 ± 0.01	0.20 ± 0.14	
a Differences between species as determined by a two-sample Wilcoxon test (α = 0.05, significant differences in bold).

Enantiomeric composition

There were significant differences between lodgepole and jack pine in the percent enantiomeric composition of limonene, α-pinene, and β-pinene (Table 4). Both populations of lodgepole pine had significantly (F = 20.75; df = 2,60; P < 0.001) higher percentage of (–)-α-pinene than the population of jack pine [and corresponding lower percentage of (+)-α-pinene (F = 20.57; df = 2,60; P < 0.001)] (Table 4). Both populations of lodgepole pine and the jack pine population had higher percentages of the (–)-enantiomer of limonene than the (+)-enantiomer. The southern BC population of lodgepole pine near Kelowna had a higher percentage of (+)-limonene than the jack pine population (F = 4.87; df = 2,50; P = 0.01), which conversely had higher percentage of (–)-limonene (F = 4.23; df = 2,50; P = 0.02). All three populations of the two species contained exclusively (–)-β-pinene (Table 4).

Table 4 Mean percentages of chiral monoterpenes (±1 SE) from lodgepole pine trees in northern and southern locations (LP-N and LP-S respectively) and jack pine trees (JP)a.

Location	(+)-α-pinene	(−)-α-pinene	(+)-β-pinene	(−)-β-pinene	(R)-(+)-limonene	(S)-(−)-limonene	
LP-S	37 (±4)	63 (±4)	0	100	29 (±5)	71 (±5)	
	n = 28	n = 28	n = 28	n = 28	n = 27	n = 27	
LP-N	50 (±6)	50 (±6)	0	100	17 (±5)	83 (±5)	
	n = 17	n = 17	n = 16	n = 16	n = 16	n = 16	
JP	78 (±4)	22 (±4)	0	100	5 (±5)	95 (±5)	
	n = 18	n = 18	n = 14	n = 14	n = 10	n = 10	
a Number of samples for each compound and location varied due to sample failures.

Induced Defenses

There were significant differences between the terpene levels sampled at day 0, 2, and 14, and between the locations (Fig. 2). There were also significant differences in the rate of change between locations for five of the terpenes: limonene, myrcene, α-pinene, β-pinene, pulegone (Fig. 3).

Figure 2 Mean levels of terpenes at day 0 (initial sample), day 2, and day 14 in two populations of lodgepole pine (LP-S – southern lodgepole pine, Kelowna; LP-N – northern lodgepole pine, Chetwynd) and one population of jack pine (JP – Fort McMurray) in response to wounding.

Different lower case letters within each terpene indicate significant differences (α = 0.05) between locations for that terpene.

Figure 3 Change in terpene levels at day 0, day 2, and day 14 in two populations of lodgepole pine (LP-S – southern lodgepole pine, Kelowna; LP-N – northern lodgepole pine, Chetwynd) and one population of jack pine (JP – Fort McMurray) in response to wounding.

Asterisks indicate significant differences in the rate of change of the monoterpene levels.

The southern BC lodgepole pine had a higher rate of increase in levels of limonene (χ2 = 14.99; df = 2; P < 0.001) between day 0 and 2 compared to the rates of increase in northern BC lodgepole pine and jack pine. The only difference (F = 4.25; df = 2, 28; P = 0.02) in limonene levels between locations was in the initial sample between the northern BC lodgepole pine and the jack pine (Fig. 2).

The northern BC lodgepole pine also had a lower rate of increase of myrcene levels (χ2 = 10.92; df = 2; P < 0.01) compared to the southern BC lodgepole pine, but neither lodgepole pine population differed from the jack pine between day 0 and 2. Only the initial sample had myrcene levels that were higher (F = 4.16; df = 2, 28; P = 0.03) in the northern BC lodgepole pine trees compared to the jack pine trees (Fig. 2).

The rate of increase of α-pinene levels in the southern BC lodgepole pine trees was higher (χ2 = 10.35; df = 2; P < 0.01) than in the northern BC lodgepole pine trees, while the rate of increase of α-pinene levels in jack pine trees was not different from either population of lodgepole pine between day 0 and 2 (Fig. 3). However, the absolute levels of α-pinene were higher (F = 20.8; df = 2, 28; P < 0.001; F = 13.68; df = 2, 28; P < 0.001; F = 45.46; df = 2, 28; P < 0.001) in jack pine on all three sample dates compared to both lodgepole pine populations (Fig. 2). The rates of increase in β-pinene levels were higher (χ2 = 12.90; df = 2; P < 0.01) in jack pine and the southern BC lodgepole pine compared to the northern BC lodgepole pine from day 0 to 2 (Fig. 3).

The rate of increase in pulegone levels was higher (χ2 = 16.93; df = 2; P < 0.001) in the northern BC lodgepole pine compared to the jack pine and the level of pulegone was higher in the northern BC lodgepole pine compared with the other two sites (Fig. 2). The level of pulegone in the southern BC lodgepole pine showed a significant rate of decrease compared with both other populations between day 0 and 2 (Fig. 3) and the level was lower (F = 19.32; df = 2,2 8; P < 0.001) than in northern BC lodgepole pine (Fig. 2). Between day 2 and 14, jack pine showed a higher (χ2 = 6.37; df = 2; P = 0.04) rate of increase in pulegone compared to the southern BC lodgepole pine (Fig. 3) although the levels at day 14 were not different between any of the locations (Fig. 2).

There was no difference between locations in rate of change in the total phloem terpene levels, although the initial sample did show higher (F = 3.75; df = 2, 28; P = 0.04) levels of total terpenes in northern BC lodgepole pine compared to the jack pine. However, the levels of total terpenes were not different at day 2 or 14.

Discussion

Comparison of bark resin terpene compositions of lodgepole and jack pine sampled in our study showed that lodgepole pine have higher constitutive levels of most terpenes than does jack pine, including 3-carene, myrcene, and terpinolene (Table 2). Many of these terpenes are important semiochemicals for the mountain pine beetle. For example, trap catches of mountain pine beetle in pheromone baited traps were enhanced by 3-carene released at a high rate (Miller & Borden, 2000). Myrcene and terpinolene together were found to be an even more effective synergist than myrcene alone in pheromone baited traps (Borden, Pureswaran & Lafontaine, 2008), while myrcene was the most effective synergist in a baited tree study (Borden et al., 1983). This suggests that lodgepole pine may be more apparent than jack pine to foraging mountain pine beetles. On the other hand we found lower concentrations of the terpenes that are generally considered to be toxic – e.g., the ovicidal limonene and 3-carene (Raffa & Berryman, 1983b) – in jack pine compared to lodgepole pine. As ∼47% of the jack pine in our study had undetectable levels of 3-carene, this host species is potentially more suitable than lodgepole pine for mountain pine beetle reproduction, although Erbilgin et al. (2013) found that jack pine released higher levels of 3-carene than lodgepole pine after beetle introduction. There is likely to be regional variation in the terpene composition of jack pine which could partially explain this difference between our observations and Erbilgin et al. (2013) just as has been observed in lodgepole pine (Forrest, 1980; Clark, Carroll & Huber, 2010). Erbilgin et al. (2013) also used cut bolts rather than living trees which could further explain the discrepancies.

When considering our results it is important to remember that there is considerable regional variation in monoterpene composition among lodgepole pine populations (Forrest, 1980; Smith, 1983; Clark, Carroll & Huber, 2010), and given the extensive historic range of the mountain pine beetle, it appears to have evolved to deal with this variation. Our sampling method does not allow us to account for any intraspecific variation in terpene levels as we are only separating by tree species. There is variation in terpene composition within the same tree species (Forrest, 1980; Clark, Carroll & Huber, 2010) but we focused on interspecific variation between the two pine species in this study.

The complex interactions between the insect and resin α-pinene levels are very important in light of the fact that levels of this monoterpene in jack pine are significantly higher than in lodgepole pine (Table 1), thereby potentially affecting the mountain pine beetle’s behavior and success in colonization of jack pine. α-Pinene (Table 1) is metabolized by female mountain pine beetle to produce trans-verbenol, the primary component of the mountain pine beetle aggregation pheromone (Conn et al., 1984), which is essential to successful mass attack (Rudinsky, 1962). Erbilgin et al. (2013) showed that the higher emissions of trans-verbenol by female mountain pine beetle on jack pine are associated with higher α-pinene levels in that host. α-Pinene is also auto-oxidized to verbenone, an anti-aggregation pheromone of the mountain pine beetle (Hunt et al., 1989), and Erbilgin et al. (2013) found equal amounts of verbenone emitted from lodgepole and jack pine bolts. Incorporating the host defenses into the chemical signals that regulate aggregation helps optimize the beetles’ success (Raffa & Berryman, 1983a).

The enantiomeric composition of α-pinene can be important to some bark beetles (Renwick, Hughes & Krull, 1976). Even though the overall levels of α-pinene were higher in the jack pine, there was a lower proportion of (–)-α-pinene compared with both populations of lodgepole pine (Table 4). Volatiles collected from bolts of lodgepole pine from southern BC showed 67.7% and 100% of (–)-α- and (–)-β-pinene respectively (Pureswaran, Gries & Borden, 2004), which corresponds well with our findings for the southern BC lodgepole pine bark and phloem (62.6% and 100% respectively) (Table 4). There is a correlation between the enantiomeric composition of α-pinene the insects were exposed to and the ratio of pheromone enantiomers produced by several species of Ips (Seybold, 1993). For example, a higher percentage of (–)-α-pinene in the host spruce [Picea abies (L.) H. Karst.] was found to correlate with a higher percentage of cis-verbenol produced by Ips typographus (Lindström et al., 1989). We found significantly higher ratios of (–)-α-pinene in jack pine compared to lodgpole pine which corresponds to findings by (Erbilgin et al., 2013). Electroantennogram studies have shown that the mountain pine beetle can differentiate between enantiomers of verbenol and verbenone (Whitehead et al., 1989), although electrophysiological responses do not necessarily translate into specific behaviors and cis-verbenol was not as attractive as trans-verbenol to mountain pine beetles when combined with myrcene and exo-brevicomin (Miller & Lafontaine, 1991). However (Erbilgin et al., 2013) did not find the production of (−)-trans-verbenol to be limited by the enantiomeric composition of α-pinene in the jack pine.

There were also differences in the rate of change in the levels of some terpenes after wounding, and if induced defensive responses are adequate and rapid beetles will sometimes abandon their colonization attempts (Raffa, 1991). Uninfested lodgepole pine trees near Kelowna, BC were hard to find due to the high level of beetle activity in that area. While the sampled trees were uninfested, likely due to chance or geographic distance from infested stands, it is also possible that they may have possessed some characteristic that made them relatively unsuitable for colonization. Furthermore, in an attempt to standardize the levels of observed induced defenses these trees were only mechanically wounded as there is large variability between the beetles and the microorganisms they carry (Lee, Kim & Breuil, 2006). Our results could therefore be different than the induced response that would be caused by beetle attack.

Limonene, a terpene toxic to bark beetles (Smith, 1965; Raffa & Berryman, 1983b; Cook & Hain, 1988), was found to have a higher rate of increase between day 0 and day 2 in the southern population of lodgepole pine compared with the northern lodgepole pine population. This stronger response by the southern lodgepole pine after wounding compared with the northern pine trees supports the hypothesis that tree populations with more prior exposure to mountain pine beetle outbreaks maintain a more effective response to attack – i.e., rapid increase in levels of a toxic terpene – and hence are also less suitable for reproduction (Cudmore et al., 2010). The rate of increase of myrcene as well as the levels and rate of increase of β-pinene, did not differ between southern lodgepole pine and jack pine, which indicates that the beetle would experience a similar rate of response in this new host species for these terpenes.

The outcome of the interaction of the mountain pine beetle with phloem resin constituents in a novel host such as jack pine could also be dependent on beetle population levels. Boone et al. (2011) found that the resin defenses in lodgepole pine play an important role in protecting trees from mountain pine beetle attack at low population levels but not at high populations. Our results suggest that due to the generally lower levels of terpene-based defenses in jack pine relative to lodgepole pine, incipient-epidemic populations of mountain pine beetles may have a greater success in colonizing jack pine. Furthermore, beetles may be able to exploit the relatively higher levels of α-pinene in jack pine to produce aggregation pheromones that further increase colonization success (Erbilgin et al., 2013) and allow populations to increase rapidly.Cullingham et al. (2012) predicted the distribution of lodgepole and jack pine using genetic information. Their work indicates that the range of pure jack pine begins further east than the historical range given by Little (1971). Based on Cullingham et al. (2012) we may have sampled trees that we have identified as jack pine in this study that are not necessarily considered pure. However, we have shown that the mountain pine beetle is moving into an area that has significant differences in the absolute and relative terpene levels regardless of the hybrid-genetic status of the hosts that they encounter. Most studies on the behavioral and toxic effects of terpenes on mountain pine beetle are conducted using lodgepole or ponderosa pine (Pinus ponderosa Dougl. ex Laws.) as model host species. The minimum levels of resin terpenes necessary to provide an adequate attractant plume from an attacked bole are not known. Therefore there could be an additive effect between a number of host kairomones (Borden et al., 1983; Conn et al., 1983), which in combination with the higher levels of total terpenes present in lodgepole pine may suggest that lodgepole pine are easier for mountain pine beetle to locate than jack pine. The terpene composition may also make the lodgepole pine easier to identify as a suitable host. For example, despite what appears to be a preferable chemical environment in a novel host, whitebark pine (Pinus albicaulis), compared to lodgepole pine, Raffa, Powell & Townsend (2013) found that mountain pine beetle did not show preference for whitebark pine in mixed stands, indicating that mountain pine beetle may be better at recognizing its traditional host in such situations. However, the combination of lower levels of some terpenes generally considered toxic to insects, and an increased concentration of α-pinene, which is a precursor of both the primary aggregation and antiaggregation pheromones, may make jack pine easier to attack and colonize. This implies a substantial potential for ongoing and increased success for the mountain pine beetle in this new geographic range and host, particularly if climatic suitability increases as predicted (Safranyik & Carroll, 2006; Safranyik et al., 2010). As the beetle spreads further into pure jack pine forests, it will be critical to conduct further research into the behavior and reproductive success of mountain pine beetle in this new host.

Analytical chemistry analyses on phloem samples were performed by Mr. Clive Dawson of the British Columbia Ministry of Forests, Lands and Natural Resources Operations.

Additional Information and Declarations

Competing Interests

Author Contributions

Dr. Huber is an Academic Editor for PeerJ.

Erin L. Clark conceived and designed the experiments, performed the experiments, analyzed the data, wrote the paper.

Caitlin Pitt performed the experiments.

Allan L. Carroll and B. Staffan Lindgren conceived and designed the experiments, wrote the paper.

Dezene Huber conceived and designed the experiments, wrote the manuscript, and contributed reagents/materials/analysis tools.

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
