# Peer review of "Comparison of lodgepole and jack pine resin chemistry: implications for range expansion by the mountain pine beetle, Dendroctonus ponderosae (Coleoptera: Curculionidae)"

_PeerJ, doi:10.7717/peerj.240_

## Round 0.1 · original submission · Minor Revisions

Please take a close look at the comments from the reviews and address these. I found both reviewers to be very supportive of your manuscript, and I hope you find their suggestions to be helpful for improving your paper. Because no significant issues were identified, I am confident the paper will be accepted after slight revision.

·

Basic reporting

The basic reporting in this paper is very straightforward. It is not lacking in any way in this regard, and is a substantive analysis of the data largely appropriate to the hypothesis.

Experimental design

The experimental design and analysis contrast a little. The results are based on comparativley few locations of the two conifers and the induced responses are due to mechanical wounding rather than on controlled beetle infestation of naive trees(admittedly risky). Thus the framewrork is merely adequate. The analytical chemistry is well conducted and reported. The statistical analysis is refreshingly straightforward.

Validity of the findings

The data are of high quality, but there is a problem with using the chemical components to independently address relative contributions for host plant attack, utilization as pheromone substrates, chemical apparency and other phenomena. I would say that this "listing" approach is quite common but does not allow one to understand why the host range is succesfully expanding. The authors appropriately recognize both regional variation and artificial wounding as being limitations in the study. Thus an overall interpretation basd on the individual discrete findings is not entirely coherent.

Thus the relative weighing of possible biological outcomes based on quantitative differences in individual compounds is ambitious and might be scaled back.

Additional comments

I would recommend a different approach to presenting the findings. How about listing compounds of interest in a table or two with quantitative data for each species presented simply with a potential benefit towards new host utilization scored as either a "+" or "-" relative to the initial host and incorporating a final column called biological effect. The relative outcomes could then be broadly summarized in a more straightforward manner and may make it is easier for the authors to limit the slightly speculative "listing" approach that is not fully satisfying to the reader.

Reviewer 2 ·

Basic reporting

The manuscript is well written and clearly presented, but with room for improvement as detailed in my general comments to the authors.

Experimental design

The experimental design is very simple, with sampling of 10 trees in each of several sampling sites along a transect through the distribution area of the mountain pine beetle. The objectives are very clear: to compare the suitability of two pine hosts of the mountain pine beetle. The methods are standard and are clearly presented.

Validity of the findings

The data are sound, but could have been better presented - see my suggestions under general comments to the authors.

Additional comments

Comparison of lodgepole pine and Jack pine resin chemistry…
by Clark, Pitt, Carroll, Lindgren, Huber

General comments
I have been looking forward to a paper on pine resistance to the mountain pine beetle in its expanding range east of the Canadian Rockies. This manuscript addresses this interesting topic and will be a welcome addition to the literature on this extremely damaging forest pest. The manuscript is nicely written, although the discussion could have been better outlined and organized.
1. Outline of discussion: you have a very clear objective in this manuscript, and that is to compare the defensive capability of lodgepole pine and jack pine. I suggest that you organize the discussion around this comparison much more explicitly than you do in the present version. Discuss and explain how the different terpene parameters you have studied make the trees more or less suitable for host colonization by the mountain pine beetle.
2. Literature: there are some recent and very relevant papers that should have been included: Raffa et al. 2013, PNAS 110: 2193-98; Erbilgin et al. in press, New Phytologist, online. See also my comment to line 72 in the introduction for other papers that could be included. In general, the reference list is a bit weak on more recent papers from let’s say the last 5 years. The references are also almost entirely addressing North American biological systems, and I could only find one reference to a European bark beetle species. The focus on North America is of course understandable given the topic of the manuscript but there are probably general insights to be gained from other bark beetle-conifer systems.

Specific comments
Abstract
Middle of page: delete “and data were compared between the two species” since this is self-evident and what you do in the next sentences.
Six lines from the bottom: The sentence starting with “Differences in wounding…” is hard to read and should be improved. The results on induced chemistry are presented in very general terms and are difficult to grasp. Yes, you found some differences within and between species, but what exactly where they, and do they suggest jack pine to be more or less resistant than lodgepole pine?
Two lines from bottom: delete “a” from “a jack pine”.

Introduction
Line 40-42: this sentence could be deleted or moved.
Line 51: add some words “i.e. they enter relatively ‘defense-free space’”.
Line 66: “be toxic to bark beetles”. You can find some newer references to back up the claim that terpenes are toxic, e.g. Raffa et al. 1985, Environ. Entomol. 14: 552-56.
Line 72: is it correct that terpenes increase in the heartwood after infection – that is new to me. Perhaps you mean phloem? When documenting the ecological roles of induced terpene defenses in conifers (line 70-75) you could also mention recent publications highlighting the importance of induced defenses for tree resistance (e.g. Schiebe et al. 2012, Oecologia 170: 183-98; Zhao et al. 2011, PlosOne 6: e26649; Boone et al. 2011, in your reference list).

Methods and materials
Line 94-95: delete the definition of breast height, as this also appears on line 90.
Line 95-98: I don’t think it is necessary to provide details on the type of envelope or ultrafreezer used. If we were to go into that level of detail it is hard to know where to stop.
Line 106: why are concentrations given as ppm? Did you not use an internal standard?
Line 110: write “Values below 5 ppm…”
Line 136-137: it sounds a bit strange to state that the trees were sampled at the time of sampling. I understand what you mean, but you should try to reformulate. You could say that the trees were sampled on August X (initial sampling) and then 2 and 14 days later.
Line 140-144: I wonder if it really is as simple as this. Induced defenses that are building up later than 2 days after an attack can perhaps kill the eggs and developing brood.
Line 150-152: I am not sure if I understand this. Did you identify and quantify 9 individual monoterpenes in the chromatograms and then calculate the total for 26 peaks, some of which you did not quantify individually? Please explain. Also, I don’t understand the meaning of “all 26 terpenes tested” – you didn’t test them but only measured their concentrations.

Results
Line 162: simplify to “We sampled a total of 50 lodgepole pine…”
Line 166 and elsewhere: it may not be clear to all readers that AB means Alberta – please spell it out.
Line 170: The passage on the next few lines would be easier to follow if you say that lodgepole pine had higher levels of 20 of the 23 monoterpenes that differed significantly between the two species, and then go on to list the three that were significantly higher in jack pine.
Line 183: delete the word “sampled”.
Line 187-214: the results on induced terpene levels could be presented clearer and be more economically written. It would be useful if you could summarize the main differences between lodgepole pine and jack pine at the beginning of this subchapter. Now you plunge straight into a rather detailed description of the results, and it is hard to see the overall picture. Perhaps you can reorganize the presentation and focus on the comparison between localities rather than listing the results compound by compound. As a reader I am more interested in getting a clear comparison of tree species/populations than of limonene, myrcene etc.

Discussion
Line 228-230: here you could refer to the recent paper by Raffa et al. (2013), who came to similar conclusions when comparing lodgepole pine and whitebark pine.
Line 230: replace “We also” with “On the other hand we…”, since this sentence presents results suggesting that jack pine is a more suitable host tree, in contrast to the results presented above, which suggest lodgepole pine to be more suitable.
Line 235: This sentence is incomplete – you must explain why and how the high levels of a-pinene are “alternative” to the preceding paragraph.
Line 238: you should probably use a stronger word than “facilitates” – perhaps “is essential for”. The primary component in the pheromone must be essential to achieve mass attacks.
Line 241-44: this sentence is too long and complex and should be divided in two. In line 241 you could rewrite to “a-pinene has also been found to be ovicidal…”.
Line 246-47: Explain the significance of this statement for the colonization success of MPB. Similarly in line 250-52 - explain how this would happen.
Line 252-57: this section is quite wordy, general and vague. Please be much more concrete and to the point.
Line 258-60: do not start the sentence with this reservation - it makes me less interested in the part that follows. Also, add the word “studies” after “electroantennograms” and replace “this insect” with “mountain pine beetle”.
Line 260-70: You must do a better job at explaining how all these specific details are relevant in your context. Delete the first part of the sentence on line 260 (“In the…instance,”) and the last part of the sentence on line 266-67 (“found in…typographus”). Lindstrøm et al. is not in the reference list.
Line 274-79: to me, these two sentences are the most important and interesting ones in the whole discussion. I suggest that you lift these up to the start of the discussion, since they nicely summarize the comparison of the two pine species.
Line 283: “it has probably evolved…”
Line 288-90: there are some newer relevant papers out there from other bark beetle-conifer systems – see my comments to line 72 above.
Line 294-97: you could have used a chemical elicitor such as methyl jasmonate to standardize the treatment.
Line 315: use the common name of D. ponderosae.

Figures and tables
Figure 1: if you increase the symbol and font size this map can be made quite small. If you replace the symbol for lodgepole pine with e.g. a black triangle it will be even more distinct from jack pine. You could also mark the sites where you sampled for induced chemistry by e.g. adding a circle around the species symbol.
Figure 2: the meaning of day 0 etc. must be explained more explicitly in the legend. You probably don’t need to provide the name of the locations in the legend. The font size in the figure panels is too small. If you want to highlight the increase in terpene levels over time at the different sites it would probably be better to switch day and site in the figure. That is, have one figure panel for each site and show data for each day as differently colored bars. As a bonus the differences between jack pine and the two lodgepole pine populations would probably become clearer.
Figure 3: same comments as for Figure 2 regarding the legend. You should remove the x-axis labels from the two upper figure panels. The figure would probably me more readable if you put the y-axis labels and legend on the right hand size of the panels to the right (showing data for a-pinene and pulegone). Also, you don’t have to repeat the y-axis legend for every single figure panel.
Table 1: this table is probably not necessary. The location of the sampling sites can be shown in Figure 1, as suggested above. The number of trees and starting dates for sampling can be given in the text (Day 2 and Day 14 dates follow from the starting date). The table legend does not contain enough information to be understandable on its own – you need to also provide the name of the tree species.
Table 2: the legend is not understandable on its own. The terpene data should only be presented with one decimal and the numbers should be aligned by decimal point to make them more readable. Why are data presented as ppm and not as absolute values (e.g. mg per g dry weight)? Perhaps you could identify which compounds are monoterpenes and which are sesquiterpenes?
Table 3: same comments as Table 2. These two tables could easily be combined into one large table.
Table 4: replace the word “compounds” in the legend with “monoterpenes” to be more specific. The meaning of LP-S etc. should be explained better (i.e. the meaning of S and N). The table would be easier to read if the column headings were modified: let the name of each compound (e.g. a-pinene) span two columns, and put (+) and (-) on a new line underneath it.

---

## Round 0.2 · accepted · Accept

Thanks for your revision, and helpful indications of how you responded to reviewers comments.